# Qualitative assessment of the impact of socioeconomic and cultural barriers on uptake and utilisation of tuberculosis diagnostic and treatment tools in East Africa: a cross-sectional study

Elizabeth F Msoka [ORCID],[1] Fred Orina,[2] Erica Samson Sanga [ORCID],[3] Barbara Miheso,[2] Simeon Mwanyonga,[3] Helen Meme,[2] Kiula Kiula,[1,4] Alphonce Liyoyo,[1] Ivan Mwebaza,[5] Augustus Aturinde,[6,7] Moses Joloba,[5] Blandina Mmbaga,[1] Evans Amukoye,[2] Nyanda Elias Ntinginya,[3] Stephen H Gillespie,[8] Wilber Sabiiti[8]

► Prepublication history and supplemental material for this paper is available online. To view these files, please visit the journal online (http://dx.doi.org/10.1136/bmjopen-2021-050911).

EFM and FO contributed equally.

For numbered affiliations see end of article.

**Correspondence to**
Dr Elizabeth F Msoka;
e.fbright@kcri.ac.tz

## ABSTRACT

**Objectives** Early diagnosis and timely treatment are key elements of a successful healthcare system. We assessed the role of socioeconomic and cultural norms in accelerating or decelerating uptake and utilisation of health technologies into policy and practice.

**Setting** Secondary and tertiary level healthcare facilities (HCFs) in three East African countries. Level of HCF was selected based on the WHO recommendation for implantation of tuberculosis (TB) molecular diagnostics.

**Participants** Using implementation of TB diagnostics as a model, we purposively selected participants (TB patients, carers, survivors, healthcare practitioners, community members, opinion leaders and policy-makers) based on their role as stakeholders. In-depth interviews, key informant interviews and focus group discussions were held to collect the data between 2016 and 2018. The data were transcribed, translated, coded and analysed by thematic-content analysis.

**Results** A total of 712 individuals participated in the study. Socioeconomic and cultural factors such as poverty, stigma and inadequate knowledge about causes of disease and available remedies, cultural beliefs were associated with low access and utilisation of diagnostic and treatment tools for TB. Poverty made people hesitate to seek formal healthcare resulting in delayed diagnosis and resorting to self-medication and cheap herbal alternatives. Fear of stigma made people hide their sickness and avoid reporting for follow-up treatment visits. Inadequate knowledge and beliefs were fertile ground for aggravated stigma and believing that diseases like TB are caused by spirits and thus cured by spiritual rituals or religious prayers. Cultural norms were also the basis of gender-based imbalance in accessing care, 'I could not go to hospital without my husband's permission', TB survivor.

**Conclusion** Our findings show that socioeconomic and cultural factors are substantial 'roadblocks' to accelerating the uptake and utilisation of diagnostic and treatment tools. Resolving these barriers should be given equal attention as is to health system barriers.

### Strengths and limitations of this study

► This is a qualitative assessment using triangulated data sources to strengthen the validity of results.
► Valuable insights into the role of socioeconomic and cultural barriers on access and utilisation of tuberculosis technologies.
► Investigators were experienced and well trained on qualitative data collection.
► Views of 712 participants may not necessarily represent views of everyone in East Africa.
► Participant's ability and/or inability to recall past events may limit the accuracy of the data collected.

## INTRODUCTION

Tuberculosis (TB) claims more than a million lives every year of which the poorest of the world are most affected.[1–4] Considerable technological advances have been made in the diagnosis and treatment of TB but they remain less accessible to the communities that need it the most.[5–7] Only two thirds of the global reported TB cases are confirmed by a laboratory test[2]

Tanzania, Uganda and Kenya are among the 30 countries with highest burden of TB as well as TB/HIV coinfection in the world.[1] The high TB burden is characterised by poor diagnostic capability where many of high burdened countries still depend on microscopy, a less sensitive tool for diagnosis.[8–11] Each of the three countries has a national TB and Leprosy control programme through which TB services are provided.[12] Diagnosis and treatment are free though the prediagnosis screening that most patients go through is not free. The length of time taken to getting proper diagnosis depends on the healthcare

worker's ability to recognise TB symptoms and person's health seeking behaviour. Although medicines are free, patient care and transports costs to and from healthcare facilities (HCFs). WHO approved Molecular diagnostics, Xpert *Mycobacterium tuberculosis*/Rifampicin (MTB/RIF) test and Line probe assay for rapid detection of TB and drug resistance that have been found to be implementable though the socioeconomic changes associated with their uptake and utilisation are less elucidated.[10 13]

Knowledge of disease, social and cultural issues have been highlighted as important factors in the patient diagnosis and treatment pathway of TB.[14] Studies conducted elsewhere regarding TB and social cultural barriers have revealed that there is a complex interplay between contextual factors and community understanding of the disease. Cultural beliefs about causality and treatment-seeking paths were often mentioned in conjunction with biomedical views.[15] There was a strong interface between TB and HIV in communities, and knowledge of TB is often confused with HIV, additionally HIV-related stigma has been extended to those living with TB.[15] The stigma of TB infection being related to HIV infection has been shown to have an impact on patients seeking treatment and their adherence to treatment.[16–18] These cultural barriers have been reported in most African communities where TB infection is associated or linked to witchcraft as responsible for leading to delay in accessing diagnosis and treatment of TB.[19–21] Interestingly, lack of access to media was highlighted as significantly associated with low TB knowledge.[22]

Socioeconomic activities such as mining are reported high risk for communicable illness including TB and HIV/AIDS.[19] Population migration has been shown to facilitate transmission of TB as well as causing treatment disruptions.[23 24] We set out to investigate these factors and the ways they impede uptake of health technologies in order to inform models for resolution in the East African region and the low-income and medium-income countries setting at large. We show that socioeconomic and cultural norms substantially limit access and utilisation to healthcare services and resolving them is fundamental to having well-functioning health systems.

## METHODS
### Approach
This was a qualitative part of a larger mixed methods study assessing health system and wider community impediments and opportunities to unlock them in order to achieve effective uptake and utilisation of diagnostic and treatment tools. Implementation of WHO approved TB diagnostics was used as benchmark to evaluate the phenomenon. The study was conducted between January and December 2017 in Uganda, Kenya and Tanzania. At the time of the study, only Kenya was defined as a middle-income country, and it has now been joined by Tanzania in this classification. This means the Tanzania results in this study reflect a low-income country setting.

### Study design, participants and researchers
This was a cross-sectional study targeting TB patients and survivors, caregivers, general members of the community served by the HCF, healthcare practitioners, opinion leaders, local government authorities and policy/decision makers at local and national levels from these three countries. An interdisciplinary team of researchers consisting of social scientists, biomedical scientists and clinicians conducted the study.

### Context
Although the study countries: Kenya, Tanzania and Uganda are high TB burden countries, the study did not aim to study TB as a disease but as a prism to look into the challenges to increasing access to health technologies. Although there are pockets of uniqueness in each country, there is substantial level of cross border interaction and similarities in political and health system administrative structures. The differences and similarities provided the ideal context to study common socioeconomic-cultural issues and how they impact access and use of diagnostic and treatment tools.

### Sampling strategy
A purposeful sampling method was used to select participants based on these categories and also to ensure representation by age, socioeconomic status, gender and geographical location. A combination of in-depth interviews (IDI) and focus group discussions (FGDs) were used to generate data for this study. Participants who participated in the IDIs were different from those who participated in the FGDs. Participants for the FGDs were identified through the community leads who acted as important gate keepers in accessing these groups of participants, as they knew the area very well, were respected and trusted by the community, and were able to help with identifying potential participants. In each selected administrative region, five FGDs were undertaken with 12 participants per group and 10 IDIs were performed. FGDs were meant to capture a collective opinion over issues concerning diagnosis and treatment of TB. IDIs were to get a deeper opinion of individuals without interference or fear of others (online supplemental material: interview guide for focus groups).

### Data management and analysis
All the IDI and key informant interviews (KIs) and FGDs were conducted by a trained qualitative data investigators who underwent a 5-day training held in Kampala, Uganda, May 2016. All the data collection tools including questionnaires and interview guides were pretested and revised for clarity and accuracy prior to being used in the field. The interviews were conducted in Swahili in Kenya and Tanzania and/or English in Uganda, in the offices or health facilities depending on the group of respondents. Audio recorders were used to record all interviews with permission from the respondents. Collected data were transcribed verbatim from Swahili and translated

to English before being analysed by thematic-content approach.[25] Data were coded and synthesised using both inductive and deductive techniques by the first author and reviewed by two other qualitative data investigators for consistency to produce list of themes, theme models and classification related to the themes.[26] In keeping with the measures advised by Barbour[27] independently, two researchers read and reread all focus group and interview data, recording the emergent themes. The process involved careful reading of the interview transcripts for pattern recognition which were later translated to categories, subcategories in order to draw the conclusions from the data meanwhile developing themes. Themes further analysed to identify subthemes. However, in the process where there was disagreement in the themes and subthemes developed, discussion took place until agreement to ensure trustworthy of the data was achieved.[27 28]

### Public involvement statement

This was a people-centred study involving people at grassroots up to policy-makers. The views presented in the paper are views of the community groups and individuals in the study countries. This qualitative paper is the voice of the people: patients, survivors, carers, opinion leaders and policy-makers. Regional and national public and policy-makers' gatherings were held to share the findings of the study and policy briefs submitted to respective ministries of health.

## RESULTS

A total of 712 participants above 18 years of age participated in this study of whom 70% were females. Sixty-five per cent were from rural areas. Table 1 highlights the characteristics of the study participants. Eight categories of individuals participated including 38 laboratory

**Table 1** Categories of respondents that participated in the study

| Respondents | Total no |
| --- | --- |
| Laboratory technicians | 38 |
| TB patients | 32 |
| TB survivors | 10 |
| TB patient care givers | 27 |
| Local government authorities | 29 |
| Medical administrators and District Tuberculosis and Leprosy Coordinators (DTLCs) | 37 |
| Medical practitioners/clinic staff/Directly Observed Therapy (DOT) nurses | 42 |
| Private practitioners | 19 |
| Key informants | 21 |
| Community members | 187 |

Community members participating in FGDs and interviews constituted the largest proportion of participants.
FGDs, focus group discussions; TB, tuberculosis.

technicians, 37 medical administrators, 29 political leaders, 42 TB patients and survivors, 36 key informants, 42 healthcare providers, 27 caregivers and 19 private practitioners in the private and public sectors. One hundred and eight FGDs were held among community members, TB patients and caregivers. A total of 270 IDI including KIs were conducted (table 1).

Participants came from 13 regions and fourteen counties with high prevalence of TB and HIV. In Tanzania, the study was conducted in the following 13 regions: Kilimanjaro, Manyara, Tanga, Mwanza, Kagera, Kigoma, Singida, Mbeya, Dodoma, Dar-es-salaam, Mtwara, Rukwa and Katavi. In Kenya, the study was conducted in the following 14 counties: Taita, Taveta, Mombasa, Garrisa, Wajir, Meru, Marsabit, Makueni, Muranga, Nairobi, West Pokot, Bomet, Bongoma and Nyamira. In Uganda, the study was conducted in the following regions and subregions including Northern region in Karamoja, Lango, Acholi and West Nile; Eastern region in Teso, Sebei, Bugisu, Bugwere, Samia, Bunyole, Jopathola and Busoga subregion of Uganda; Southern region in Buganda, Buuruuli and Kooki subregion of Uganda; Western region in Bunyoro, Rwenzori, Ankole and Kigezi subregion of Uganda (figure 1).

### Interview and FGD outcomes

A total of 712 interviews and FGD transcripts were obtained. Following analysis, the responses were grouped into two thematic categories: sociocultural and socioeconomic barriers, each containing subthemes including: knowledge of TB, diagnosis and treatment, size, taste and number of pills and side effects of TB treatment and work patterns and responsibilities.

### Sociocultural barriers
#### Beliefs and use of complementary medicine
Several beliefs about TB disease, its causes and treatment existed in the community that delayed community members to seek care at health facilities. Regarding the cause of TB, some participants believed that it is caused by witchcraft. For example, one TB patient said,

> '…Before I was diagnosed to have this disease, I had consulted more than one traditional healer. My mother believed that I was bewitched' (TB patient, 3 TZ).

In addition to cultural beliefs about the cause of TB, a variety of beliefs about alternative medicines of TB in the community were widespread as reported by the study participants. The community members did believe that the readily available local medicines in the community have the capacity of curing TB. For example, one TB patient said, 'here is a medicine called Irubukoi and most of the people use them to cure TB and pneumonia" (TB Patient, 4 TZ]

Apart from the local medicine and traditional healers, some community members believe TB is caused by spirits and can be treated through prayers in churches or mosques as illustrated below:

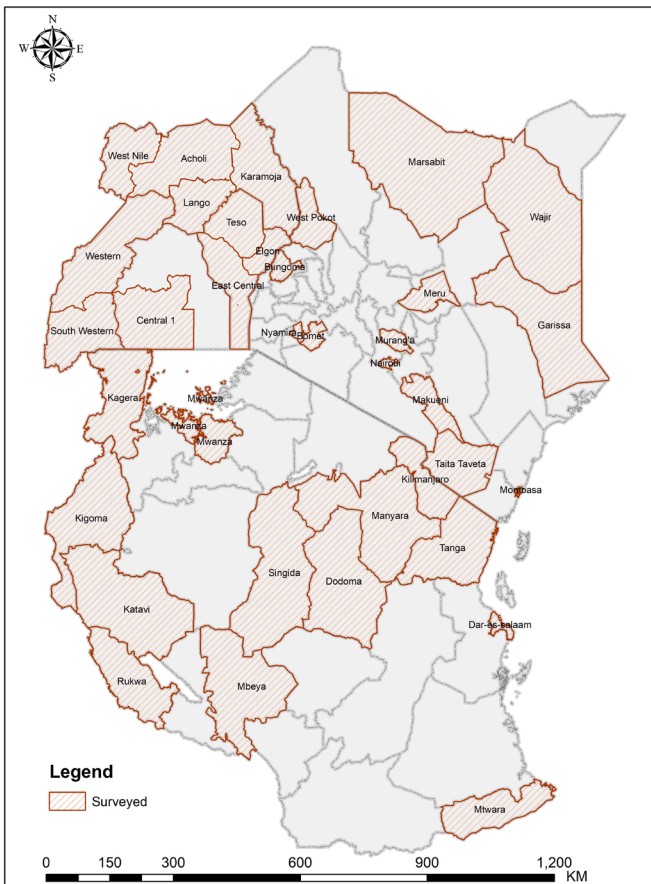

**Figure 1** The map showing the surveyed regions and counties in three East African Community Countries.

'…There is this community of Nubis, they are our relatives, and we intermarry a lot, but they love their things to do with spirits. They believe a sick person is possessed by spirit and hence they go to their spiritual leaders for prayers, yet they have TB or HIV…' [TB Care Giver, 3 KE]

The beliefs which make people think of other ailments first when they are confronted with TB symptoms and/or make them delay seeking treatment from HCFs or not seek it at all are quite prevalent among communities in the region. One healthcare provider shared example of how such beliefs manifest: 'we meet some clients who had non-medical treatment from home, because of the culture. There are people who tried to use herbs. They tried to treat a number of conditions with herbs. And they tend to take too long to seek treatment from HCFs. An attitude like: 'this one is culturally known to do it, even if you go for the medical treatment, it may not help, let's use this' [Health care provider, 1 KE].

Another healthcare provider said 'everyone knows about this, even children. Their mothers first buy them syrups over the counter not prescribed by the doctor and they try treatment at home. When their treatment fails or seems to be failing that is when they think of going to the clinic. When we ask what treatment they took from home, people often buy first-line antibiotic treatment-someone

can buy amoxicillin, if there is no improvement, they change to another antibiotic of different name thinking that there will be improvement thereafter.' [Health care provider, 2 TZ*]*

### Knowledge of TB, diagnosis and treatment

Knowledge about TB disease, causes, recognition and interpretation of signs and symptoms, and treatment was found to be low in the community members with about 50% of respondents having some basic knowledge of TB. The low knowledge of TB was more pronounced rural communities compared with their counterparts living in the urban areas. The statements below illustrate the low knowledge on TB disease.

'…TB is transmitted by doing heavy jobs like farming and engaging in sexual intercourse with girls and smoking…' [TB Patient, 1 TZ].

In contrast to the general community members, people who have suffered from TB and caregivers of TB patients had high awareness of TB and basic knowledge of TB disease. The majority of TB patients and their caregivers knew basic facts about TB transmission, common signs and symptoms, and that TB is curable. In addition, some even mentioned about the association between TB and HIV. For example, one interviewee said:

'…TB is a chronic illness. TB symptoms are severe cough of more than two weeks, coughing the sputum which are blood mixed, fever during the night and overnight sweating." [TB Survivor, 2 TZ].

Despite the high awareness of TB among TB patients and their caregivers, it was noted that most TB patients and caregivers were not aware of TB symptoms before they had suffered from the disease. It was only after diagnosis and getting education from the healthcare providers that they got to know some basics of TB as one TB survivor commented,

'…I really never thought it could be TB because I was coughing, and I thought it was a normal cough or something like pneumonia. TB is not a small disease' [TB Survivor, 3 TZ].

As a result of low awareness of TB disease, recognition and interpretation of TB symptoms and signs among community members, perceived symptom severity was what pushed them as last resort to take their patients to the health facility for diagnosis and treatment, as stated below:

'…First the (Tb patient) started coughing and to lost weight, we saw his body changing daily, he lost appetite, he was tired all the time, very much sweating especially at night he was complaining about chest pain. From there his condition continued to be worse, we decided to take him to hospital…' [Male Care Giver, 1 KE].

Another caregiver alluded to this notion by stating

'…my son experienced regular fever and coughing for almost four weeks then we brought him to the hospital. But we did not notice its TB'. We Tanzanians have a bad habit of taking things easy. We thought it was a normal cough but after four weeks…he was getting very serious, that is when we brought him to the hospital where we found it is TB' [Female Care Giver, 2 TZ].

### Size, taste and number of pills and side effects of TB treatment

Apart from complementary medicine, beliefs and traditions, the number and size of pills was among the challenges mentioned to interfere with treatment adherence as reported by a male caregiver '…the huge size of these pills sometimes discourages patients from swallowing them…when I used to give the pills to my father he used to complain that they were too big to swallow and thus he wanted the small ones.'[Male care giver, 4 TZ] Another interviewee mentioned '…the moment you are discovered remember you go, you undergo these 4 tablets plus the other ones 5, plus if you are co-infected more, so its pill burden…' [TB patient,5 UG]. In a focus group discussion, a female survivor of TB [4 TZ] pointed out bitterness of the pills as another hindrance to taking the medicines and suggested sugar-coating the pills to make them more palatable.

The side effects associated with anti-TB drugs were pointed out as one of the major reasons as to why TB patients apparently fail to complete treatment. Drawing on their lived experience, respondents shared descriptions of the side effects of the drugs '…All of them (patients on treatment) will complain of dizziness. Others have vomiting if they are not getting used to the medicine. Others reacted to the medicine and lost hearing. What we call neuropathy, the nerves, they will get a lot of pain-you get up in the morning and you are unable to walk. When you step on the ground you feel like you are stepping on thorns and sharp objects.' [Health care provider, 3 TZ].

Another healthcare provider added by saying '…Somebody treated, declared cured, comes back with a cough. About three of them, all of them I managed to send the samples to Mulago for what we call mycology culture. All of them have shown to be having fungal diseases, which is making them weak and when they are coughing this time they don't cough like TB. The cough is severe. They get wasted in a shorter time. That fungus it is not TB. Well, the drug you use for treating TB are very difficult drug, in terms of their compromising the immunity. Now, once immunity drop to a certain point and remember the lung is damaged. And still this fungus also takes advantage of the weak immunity and the damaged lung to develop in your body.' [Healthcare provider, 4 UG]

### Stigma associated with TB disease

The study found that stigma was rife among the patients, family, relatives, friends and neighbours, including perceived and enacted stigma as well as discrimination.

All TB patients interviewed reported experiencing stigma and discrimination from the immediate family members, the community and even at health facilities. The following statements illustrate how TB patients were stigmatised and discriminated

'…I used to be fearful; ashamed I could not move a lot because I knew people were talking about me. I stayed in the house. I would just get outside next to my house and warm myself in the sun because I did not have visitors, they did not want to come…' [TB patient, 7 KE].

In addition, TB patients are also stigmatised to having HIV too. There was a sense in the community that if one has TB, s/he is most likely to have HIV too. Talking about this misconception, one participant said '…most of the people know that when a person is having TB then this person is likely to have HIV/AIDS as well…[Community member, 1 TZ]". Another one said '…because they appear emaciated, people might think they have HIV but some may not be having HIV they are just TB without other comorbidity…So I think people should not be afraid the tests… ' [Private practitioner, 1 TZ]

### Fear of HIV testing

The policy recommending HIV test for every TB patient together with the perception that TB patient must have HIV, has created fear among people fear to go for TB testing. The community members fear that they will be asked to check for HIV status as well which eventually delays access and use of the TB diagnostic services. The following statements illustrate the fear of HIV testing in the community.

'…People fear to go to the hospital for TB diagnosis because they will be diagnosed and if they have TB, HIV will also be tested …' [TB Survivor, 6 TZ]

### Socioeconomic factors
### Costs related to health facilities

A common view among most of respondents especially in the rural settings was that costs incurred in using health services including transportation was a significant socioeconomic barrier to accessing TB diagnosis and treatment services. For example, one TB patient said '… when you plan to come to the hospital you need to arrange for bus fair, accommodation cost. You cannot come to town and go back home the same day… It is very far' [Female TB Patient, 13 TZ]. This view was echoed by another informant who said '…I use Tz.shs. 2,000/=to pay for the bus transport coming to hospital and returning home…' [Female Care Giver, 5 TZ].

'…Sputum analysis you pay Ug.shs. 20,000/= for the gram stain that is when you are expecting a bacterial infection: and the ZN (Ziehl-Neelsen) that is now testing for the AFBs (Acid-Fast Bacilli), which is for TB, which causes TB. So, the whole test cost Ug.shs.

20,000/= Ug.shs. 10,000/= for the gram stain and Ug. shs.10,000/= for the ZN.' [TB patient, 9 UG]

Despite the costs, geographical accessibility also played a significant role in hindering community members to access to health facilities for TB diagnosis and treatment adherence. One caregiver commented '...so from here to our hospital is a bit far since we are in the interior and especially when we have rains like this time, to travel from here to the hospital is very difficult because accessibility of roads is very hard...' [Care giver, 6 TZ].

It is the general view of respondents that their respective HCFs are ill-equipped for TB care, as indicated by shortage of drugs to treat TB, especially Multi-Drug Resistant TB (MDR-TB) and they have to refer patients. A respondent, for example, testified:

'... In case we don't have medicine at that time for MDR-TB you would have to transport the patient to Mulago. The problem is appropriate vehicle to transport them as they couldn't travel in the public bus. We must get a pickup (truck) for them to sit and refer them. We usually refer to the national TB centre. But that is not encouraged, because you are now spreading a dangerous disease. It is easier to bring the medicine-source nearer than sending the person *farther afield to collect the medicine.'* [Healthcare provider, 5 UG].

### Lack of food during treatment

Shortage of food was one the socio-economic concerns highlighted for non-adherence to TB medication. This theme recurred throughout the data set as a significant socioeconomic barrier to keeping up with medication adherence. Due to lack of food, some patients skipped medication or even stopped medication. For example, one TB patient said

'...there are few times when I skipped my dose if I did not have enough food to eat, because if you take those drugs on an empty stomach, they make you feel very bad for like two three hours'. [TB Patient, 10 KE] Another TB patient also echoed this view, '...TB medicine is a dose of six months, and you should take it every day. Others do stop using the medicines because of lack of food" [TB patient, 11 TZ].

### Work patterns and responsibilities

Working patterns and conditions such as mining not only creates a high-risk environment for TB transmission but also prevented miners from accessing TB diagnosis as well as treatment adherence. For example, one miner comment*ed* '...The young men working in mining areas because we are at a very high risk of contracting TB but when we fall sick, our bosses do not give us anything...' [Community member, 2 TZ]. Apart from working patterns and conditions, family responsibilities among caregivers somehow do play role in hindering caregivers' accessing TB diagnosis services and even keeping up with

TB treatment adherence. This notion was shared by one village leader who stated '...most of the TB patients are poor, so when they feel relief even in the middle of the dose, they tend to stop and get back to work in order to earn a living...' [Village leader, 3 TZ].

### Lifestyle

The nomadic lifestyle among the Maasai tribe also poses as a socioeconomic barrier to seeking TB diagnosis and treatments services at health facilities. Moving from one place to another greatly interferes with medication adherence. '...Maasai are living very far from the health centres because they tend to move from place to place far from town looking for the pasture for their animals...' [TB Care Giver,8 TZ]. Alcohol consumption among the young men in the societies have also shown to be a contributing factor to non-adherence with TB medications. For example, one miner commented '...most of the young men in this street are alcohol drinkers and this tendency disturbs the adherence to TB medicines..." [Community member, Miner, 3 TZ].

## DISCUSSION

The study aimed to assess and elucidate how the socioeconomic and cultural factors affect the uptake and utilisation of diagnostic and treatment tools for TB in East Africa. The study has brought to the fore the views of a diverse of group of stakeholders and unravels the similarities in socioeconomic-cultural perspectives across national borders. Most importantly, the study findings reveal the criticality of understanding the sociocultural context of service users in order for health systems to offer effective healthcare. including patients, survivors, caregivers, healthcare workers and administrators, and policymakers. Socioeconomic and cultural issues highlighted by the study: knowledge of disease signs and causes, perverted beliefs and use of complementary medicine, stigma and deterrent costs of care are fundamental and can be generalised to other disease conditions. For instance, anecdotal information shows COVID-19 has caused rise in stigmatisation of cough and use of alternative medicines. The recently reported decline in TB notification could be reflecting people fearing to seek formal treatment for cough and resorting to purchasing medicines over-the-counter or using traditional herbs[29]

Lack of knowledge or correct information about TB transmission and treatment was among the prominent factors hindering timely and successful diagnosis and treatment of TB. These findings concur with what other studies have found on this topic.[14 19–22 30 31] Shortage of communication channels like radios, televisions (TVs), and internet exacerbate lack of awareness of TB and existing medical remedies among people mostly in rural areas and mining industry. Poor or no electricity supply complicate the use of TVs and computers. Misinformation which is currently rampant in the COVID-19 era has always been a hinderance to uptake medical interventions

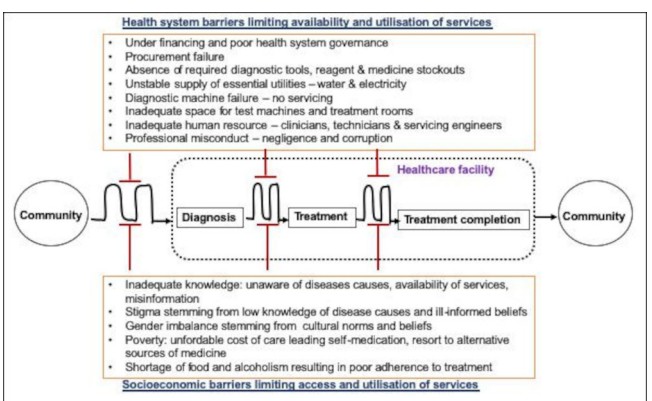

Health system barriers limiting availability and utilisation of services
- Under financing and poor health system governance
- Procurement failure
- Absence of required diagnostic tools, reagent & medicine stockouts
- Unstable supply of essential utilities – water & electricity
- Diagnostic machine failure – no servicing
- Inadequate space for test machines and treatment rooms
- Inadequate human resource – clinicians, technicians & servicing engineers
- Professional misconduct – negligence and corruption

Inadequate knowledge: unaware of diseases causes, availability of services, misinformation
- Stigma stemming from low knowledge of disease causes and ill-informed beliefs
- Gender imbalance stemming from cultural norms and beliefs
- Poverty: unfordable cost of care leading self-medication, resort to alternative sources of medicine
- Shortage of food and alcoholism resulting in poor adherence to treatment
Socioeconomic barriers limiting access and utilisation of services

**Figure 2** Diagrammatic representation of the impediments of access, uptake and utilisation of diagnostic and treatment tools in the patient pathway. Most hinderance is experienced at the primary level where the patient first seeks healthcare intervention during their ailment. The path is not straight reflecting the different barriers and routes the patient goes through to get or adhere to care.

and can only be counteracted by proactive sharing of facts.

Access and use of health technologies is further compromised by poor transport infrastructure, high illiteracy rate and lack of well-equipped HCFs and specialised healthcare providers.[32 33] Analysis of the health system barriers (first part of this study) points to the centralised implementation model as the most cost-effective of ensuring maximal uptake and utilisation of high-tech diagnostic technologies. However, this can only be effective if the governments invested in good transport infrastructure connecting rural and urban areas to ensure timely sample referral and delivery of diagnostic samples. Investing in information communication technologies would not only ensure higher awareness of TB and its care services but also communicating laboratory results to patients once tests are completed. Hiring and retaining well trained healthcare practitioners and giving them refresher training on novel health technologies cannot be more emphasised.

The current TB treatment regimen is a combination of four antibiotics for 6 months in drug sensitive TB and more months for drug resistant TB.[34] Presence of comorbidity like HIV worsens the pill burden. The pill burden (size and number of pills) was highlighted by participants as major hinderance to adherence to treatment. Side effects such as vomiting, and loss of hearing aggravated the problem. This calls for medicines developers to find innovative ways of combining drugs into single pills and less bitter to increase palatability. Finding fewer toxic drugs or combinations of drugs should be the main agenda of drug manufacturers and clinical trials. Increasing patient–practitioner communication and interaction has been suggested as one of the ways to manage and mitigate side effects caused by medicines and ensuring treatment adherences.[35] Patients should be well informed about the

side effects, pill burden as well as supportive nutritional foods they need while on treatment.

Our study found stigma to TB infection was mostly associated with HIV infection, since the general community perception is that anyone testing positive for TB is also HIV infected. As much as the perception is incorrect because HIV-TB coinfection is estimated at 40% in East Africa, it shapes the attitudes in the negative direction.[36] This contrasts the TB stigma in West Africa where TB is associated with ethnic or family curse, being grossly unhygienic and very contagious so much that even healthcare practitioners shun working with TB patients.[37] Consequently, stigma make people more inclined to keeping their ailment a secret and reduces willingness to seek medical care. This in turn stops or delays access to appropriate diagnostic and treatment services.[17 18 38 39] We believe poor understanding of the causes and signs of disease like TB creates a knowledge gap that people tend to fill with superstition and stigma. Those purposed to have knowledge such as healthcare practitioners, their stigma may be driven by lack of personal protective equipment (PPE) for which they ensure safety by avoiding patients with contagious diseases like TB. Sensitisation of both community and healthcare practitioners coupled with provision of appropriate PPE would go a long way in resolving the stigma issue.

Beliefs and myths about the causes and transmission for TB including being related to witchcraft, family curse and having a long Uvula (known as Kilimi in Swahili both in TZ and KE) as cause of recurrent coughing raised stigma and discrimination of TB patients. As a result, the patient pathway to care is complicated as most patients resort to seeking remedy from traditional healers, palatine uvula cutters before going to the formal healthcare centres. These kinds of behaviours and beliefs have been reported in other studies in done in Tanzania[38] and in other African countries.[40] Action follows belief and so someone who believes witchcraft is the cause of TB will by default consult a traditional healer or someone with spiritual powers first, which prolongs the pathway to diagnosis and treatment at formal HCFs.[35 38]

Our study is the first large qualitative study combining information from three Eastern Africa countries. It has highlighted the critical impact socioeconomic and cultural barriers have on uptake and utilisation of diagnostic and treatment technologies. Drawing from the findings it appears the largest 'roadblock' to uptake and utilisation of diagnostic and treatment tools occurs at the primary point of access in the patient pathway. This is the point between feeling poorly and getting diagnosis. It therefore makes more sense for most investment to be made at this primary point of access in order to shorten the path to access of healthcare services, which in turn would increase uptake, and utilisation of health technologies (figure 2).

### Strengths and limitations

This study brought together diverse group of stakeholders ranging from community members at village level to

practitioners and policy makers at regional and national level. Their views give an enriched perspective of how socioeconomic and cultural barriers affect uptake and utilisation of health technologies within the health system. Triangulation of data sources strengthens the validity of our results. The findings represent voices of real people and can be generalised to other diseases in as far as healthcare access is concerned. We note that views of 712 participants may not be representative of everyone in the study countries. Reliance of participant's ability to recall events of their previous interactions with the healthcare system raises risk of inaccuracy in the data collected. Quite importantly, however, is that our interviewers were well trained with experience in conducting social science studies.

## CONCLUSION AND RECOMMENDATIONS

Our study has shown that socioeconomic and cultural factors are important barriers to uptake and utilisation of diagnostic and treatment tools. While our study was done in the context of TB, we believe these challenges are fundamental and can apply to any other disease. Resolving awareness/knowledge gaps, beliefs, stigma and improving socioeconomic welfare of affected communities is an antidote that cuts across many disease conditions. We recommend future socioeconomic-cultural interventional studies with monitoring and evaluation component to assess and quantify impact of the interventions. Learning from end users by diagnostics and medicines developers will enable them to develop effective diagnostic and treatment tools. Social protection programmes to people working in high-risk industries like mining and interventions to reduce use of dirty sources of energy like firewood, charcoal, coal etc are critical for reducing the burden of respiratory diseases like TB. Focusing investment at the primary point of healthcare access is critical to maximising uptake and utilisation of diagnostic and treatment tools.

**Author affiliations**
[1] Kimanjaro Clinical Research Institute - Kilimanjaro Christian Medical University College, Kilimanjaro Christian Medical Centre, Moshi, United Republic of Tanzania
[2] Centre for Respiratory Diseases Research, Kenya Medical Research Institute, Nairobi, Kenya
[3] Mbeya Medical Research Centre, National Institute for Medical Research (NIMR), Mbeya, United Republic of Tanzania
[4] Department of Rural-Urban Development, The University of Dodoma, Dodoma, United Republic of Tanzania
[5] Department of Immunology and Molecular Biology, School of Biomedical Sciences, College of Health Sciences, Makerere University, Kampala, Uganda
[6] Department of Physical Geography and Ecosystem Science, Lund University, Lund, Sweden
[7] Department of Lands and Architectural Studies, Kyambogo University, Kampala, Uganda
[8] Division of Infection and Global Health, School of Medicine, University of St Andrews, St Andrews, UK

**Correction notice** This article has been corrected since it was published. Affiliation of Dr. Augustus Aturinde has been updated.

**Contributors** Study design: WS, SHG, NEN, EA, BMm and MJ. Data collection and collation: EFM, FO, ESS, BMi, SM, HM, KK, AL and IM. Drawings: AA and WS; Data analysis: EFM, FO, ESS, HM. Manuscript drafting: EFM, ESS, FO and WS; Manuscript review: all coauthors.

**Funding** The study was funded by the European and Developing Countries Clinical Trials Partnership (EDCTP), grant TWENDE-EDCTP-CSA-2014-283.

**Map disclaimer** The depiction of boundaries on the map(s) in this article does not imply the expression of any opinion whatsoever on the part of BMJ (or any member of its group) concerning the legal status of any country, territory, jurisdiction or area or of its authorities. The map(s) are provided without any warranty of any kind, either express or implied.

**Competing interests** None declared.

**Patient consent for publication** Not required.

**Ethics approval** The study aimed to collect views and opinions of individuals or groups of people in line with the ethical guidelines. To this end research team gained ethical approval from respective institutional and national ethics committees represented in the consortium. In the UK, approval was obtained from University Teaching and Research Ethics Committee of University of St Andrews (MD12073), Uganda: Makerere University Institutional Research Ethics Board (IRB) and National Council of Science and Technology (HS 2129), Kenya: KEMRI Scientific and Ethics Review Unit (KEMRI/RES/7/3/1) and Tanzania: KCRI and Mbeya and southern highlands zonal IRBs and National Health Research Ethics Committee (NatHREC) at NIMR headquarters (NIMR/HQ/R.8a/vol.IX/1317). In addition to ethics approvals, access permissions were sought and obtained from local government authorities and participants involved in the study gave a written consent to participate.

**Provenance and peer review** Not commissioned; externally peer reviewed.

**Data availability statement** Data are available on reasonable request. Extra data are available on request and meeting the confidentiality and use of data requirements signed between authors and participants. Email ws31@st-andrews.ac.uk of University of St Andrews to register your request.

**ORCID iDs**
Elizabeth F Msoka http://orcid.org/0000-0002-2352-3520
Erica Samson Sanga http://orcid.org/0000-0002-1764-545X

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
