## [Reviewer comments · BMJ Open]

ARTICLE DETAILS

TITLE (PROVISIONAL)	Qualitative assessment of the impact of socioeconomic and cultural barriers on uptake and utilisation of tuberculosis diagnostic and treatment tools in East Africa: A cross-sectional study
AUTHORS	Msoka, Elizabeth; Orina, Fred; Sanga, Erica; Miheso, Barbara; Mwanyonga, Simeon; Meme, Helen; Kiula, Kiula; Liyoyo, Alphonse; Mwebaza, Ivan; Aturinde, Augustus; Joloba, Moses; mmbaga, blandina; Amukoye, Evans; Ntinginya, Nyanda; Gillespie, Stephen; Sabiiti, Wilber

VERSION 1 – REVIEW

REVIEWER	Mao, Wenhui Duke Global Health Institute
REVIEW RETURNED	08-Apr-2021

GENERAL COMMENTS	This study interviewed stakeholders of TB to understand the role of socioeconomic and cultural factors in accessing TB diagnostic tools. Title: since this study completely focused on TB, it will be good to be specific in the title. Summary: the last point, the sample size (712) of this study is sufficient, as a qualitative study, if designed properly. I won't regard it as limitation. However, this study bears the limitation of interviews that the selected interviewees may not be representative for the general population. Introduction: There is abundant evidence on the impact of socioeconomic and cultural factors on service use. What is the unique perspective of this study and what is the additional contribution? For example, TB is an infectious disease and if not diagnosed and treated timely, it will spread. Therefore, understanding the socioeconomic factors that might block the service use is essential. It will be great to provide brief content of the health service provision, TB in particular, in the selected countries. Methods: It is not clear how the study participants were selected. Were they come from selected cities/countries of the three countries? Have this study covered the different geographic regions, economic development levels, and different TB prevalence of the three countries? For FGDs, please clarify if community members, practitioners & policymakers were discussed in same group?
---

	If possible, please share the interview guide/questions as Appendix. Results: Based on limited information about how questions were asked during the interviews and FGDs, it is not clear to me if the associations between socioeconomic, culture factors and TB diagnosis and treatment. Most of the results were presented as stakeholders' perspective on TB knowledge, treatment, etc. please consider to re-structure the results to reflect the associations, or position this article as knowledge, attitude, and practices of TB stakeholders. Discussion: The discussion session has expanded the findings and their implications. It will be better to compare findings from other similar studies, such as similar study in other regions, or even for another disease.
--	--

REVIEWER	Torres, Neusa Instituto Superior de Ciências de Saúde, Research
REVIEW RETURNED	14-Apr-2021

GENERAL COMMENTS	Dear authors Congratulations! Please note: pag 3 line 53/54 I suggest writing the months in full. Your abstract has long and short sentreces as well as your introduction section. Please try uniformize the size of your paragrhaps. Results I appreciate the effort to bring more quotations but I suggest choosing one more significant quotation per result. I appreciate the way you present your results in themes, it enable easy understading. All the best.
--

VERSION 1 – AUTHOR RESPONSE

Reviewer: 1	
Title: since this study completely focused on TB, it will be good to be specific in the title.	Thank you for the observation. The Title has been revised and now reads as follows: Socioeconomic and cultural barriers on uptake and utilisation of TB diagnostics and treatment in East Africa- A qualitative study. Please find the responses on page1 of the paper 12 of the paper. Thank you for the observation on the summary section regarding sample size. The section has been reviewed and the sample size of 712 has been

Summary: the last point, the sample size (712) of this study is sufficient, as a qualitative study, if designed properly. I won't regard it as limitation. However, this study bears the limitation of interviews that the selected interviewees may not be representative for the general population.	regarded as sufficient. However, the selection of the participants has been regarded as limitation of the study. Please find the responses on page 12 of the paper.
Introduction: There is abundant evidence on the impact of socioeconomic and cultural factors on service use. What is the unique perspective of this study and what is the additional contribution? For example, TB is an infectious disease and if no diagnosed and treated timely, it will spread. Therefore, understanding the socioeconomic factors that might block the service use is essential.	Thank you for the comment. The introduction section has been reviewed and the following has been added: ... Each of the three countries has a national TB and Leprosy control programme (NTLP) through which TB services are provided. Diagnosis and treatment are free though the pre-diagnosis screening that most patients go through is not free. The length of time taken to getting proper diagnosis depends on the healthcare worker's ability to recognise TB symptoms and person's health seeking behaviour. Although medicines are free, patient care and transports costs to and from healthcare facilities. Please find the responses on page 3 of the paper.

It will be great to provide brief content of the health service provision, TB in particular, in the selected countries.	
Methods: It is not clear how the study participants were selected. Were they come from selected cities/countries of the three countries? Have this study covered the different geographic regions, economic development levels, and different TB prevalence of the three countries? For FGDs, please clarify if community members, practitioners & policymakers were discussed in same group? If possible, please share the interview guide/questions as Appendix.	Thank you for your comment. The method section has been reviewed and it reads: This was a cross sectional study targeting TB- patients and survivors, care givers, general members of the community served by the healthcare facility (HCF), healthcare practitioners, opinion leaders, local government authorities and policy/decision makers at local and national levels from these three countries. A purposeful sampling method was used to select participants based on these categories and also to ensure representation by age, socio-economic status, gender and geographical location. A combination of in-depth interviews (IDI) and focus group discussions (FGDs) were used to generate data for this study. Participants who participated in the in-depth interviews were different from those who participated in the FGDs. The responses are also found from page 4 of the paper. Interview guide for FGD and In-depth interview are attached as appendix on page 17 of the paper.
Results: Based on limited information about how questions were asked	Thank you for the comment. The result section has been re-structured as follows:

during the interviews and FGDs, it is not clear to me if the associations between socioeconomic, culture factors and TB diagnosis and treatment. Most of the results were presented as stakeholders' perspective on TB knowledge, treatment, etc. please consider to re-structure the results to reflect the associations, or position this article as knowledge, attitude, and practices of TB stakeholders.	...Socio-cultural barriers Beliefs and use of alternative medicine: Several beliefs about TB disease, its causes and treatment existed in the community that delayed community members to seek care at health facilities. Regarding the cause of TB, some participants believed that it is caused by witchcraft. For example, one TB patient said, "...Before I was diagnosed to have this disease, I had consulted more than one traditional healer. My mother believed that, I was bewitched" (TB patient, 3 TZ). In other cases, the participants believed that TB is caused by long uvula as illustrated by one village leader "...Most of the people know that TB is a disease which affects the throat (KILIMI) [Uvula] and most of them go to traditional healers to remove/cut them thinking that it is the reason why they have severe cough ". [Village leader, 1 TZ] Please find the section on page 6-12 of the paper
Discussion: The discussion session has expanded the findings and their implications. It will be better to compare findings from other similar studies, such as similar study in other regions, or even for another disease.	Thank you for the comment. Discussion section has been revised as follows to highlight the value of our findings and to give a comparative context from other studies. Please find the responses from page 12-14
Reviewer: 2	
Dr. Neusa Torres, Institution Superior de Ciências de Saúde, University	

of Kwa Zulu Natal Comments to the Author: Dear authors Congratulations Please note: page 3 line 53/54 I suggest writing the months in full. Your abstract has long and short sentences as well as your introduction section. Please try uniformize the size of your paragrhaps	Thank you for the comment.
Results I appreciate the effort to bring more quotations but I suggest choosing one more significant quotation per result. I appreciate the way you present your results in themes, it enable easy understanding. All the best.	Thank you for the comment, we real appreciate it. Result section has been reviewed as follows: ...Beliefs and use of alternative medicine: Several beliefs about TB disease, its causes and treatment existed in the community that delayed community members to seek care at health facilities. Regarding the cause of TB, some participants believed that it is caused by witchcraft. For example, one TB patient said, "...Before I was diagnosed to have this disease, I had consulted more than one traditional healer. My mother believed that, I was bewitched" (TB patient, 3 TZ). ..... Please find responses on page7-12 of the paper

Title: "Impact of socio-economic and cultural barriers on uptake and utilisation of diagnostic and treatment tools into policy and practice in East Africa- A qualitative study"

VERSION 2 – REVIEW

REVIEWER	Mao, Wenhui Duke Global Health Institute
REVIEW RETURNED	22-May-2021
GENERAL COMMENTS	Thanks for revising the manuscript, I have no comments.